# An Image Style Diversified Synthesis Method Based on Generative Adversarial Networks

Zujian Yang  and Zhao Qiu *

School of Computer Science and Technology, Hainan University, Haikou 570228, China; 20081200210006@hainanu.edu.cn
* Correspondence: qiuzhao@hainanu.edu.cn

**Abstract:** Existing research shows that there are many mature methods for image conversion in different fields. However, when the existing methods deal with images in multiple image domains, the robustness and scalability of images are often limited. We propose a novel and scalable approach, using a generative adversarial networks (GANs) model that can transform images across multiple domains, to address the above limitations. Our model can be trained on image datasets with different domains in a single network, with the ability to translate images and the ability to flexibly translate input images to any desired target domain. Our model is mainly composed of a generator, discriminator, style encoder, and a mapping network. The datasets use the celebrity face dataset CelebA-HQ and the animal face dataset AFHQ, and the evaluation criteria use FID and LPIPS to evaluate the images generated by the model. Experiments show that our model can generate a rich variety of high-quality images, and there is still some room for improvement.

**Keywords:** generative adversarial networks; multiple domains; translate images

## 1. Introduction

Diversifying the style of an image essentially edits the attributes of the image, so that the image has the attributes that people require to meet their different needs. Diversifying image attributes can also increase the diversity of data, resulting in more datasets. Compiling the properties of images is a challenging problem for vision applications. Generative adversarial networks (GANs) [1] can be an important tool for generating images of people's desired attributes. After the use of GANs, the task of compiling and generating images has been greatly improved. There are applications from text to images [2–5]; unsupervised image-to-image compilation of two different domains [6,7]; and multi-domain image compilation [8], etc.

We design a model where the generator takes the input image as conditional input, and the style encoding of the target domain as the label input, and transforms the input image to the target domain indicated by the input label, where the style encoding is provided by the mapping network or the style encoder. The mapping network outputs the style code corresponding to the target image domain by randomly sampling the latent vector z and domain y. The style encoder can output the input image x and the corresponding domain into the style code s corresponding to the target domain. The mapping network consists of multi-layer perceptions (MLPs) and has multiple branches outputting, each of which can generate a style code for a specific image domain. In addition, the style encoding network consists of residual modules, MLPs and convolutional layers, and it also has multiple outputting branches, consisting of multi-layer perceptions. Style encoders and mapping networks benefit from a multi-task learning setting that can generate diverse style codes.

Our generator encodes the input picture and style, and then goes through a downsampling module, an intermediate layer module, and an upsampling module, all of which consist of residual units and convolutional attention units. The residual unit of the upsampling module contains adaptive instance normalization (AdaIN) [9]. The style encoding is

combined with the input image through AdaIN, and the scale and shift vectors are provided by learning affine transformation. The residual module is followed by a convolutional attention module to enhance the effective features in the output feature map, while some irrelevant noises are suppressed. Repeatedly superimposing the attention module can gradually improve the expressive ability of the network.

## 2. Related Work

### 2.1. Generative Adversarial Networks

The generative adversarial network (GAN) model was originally proposed to generate images from random noise. Its structure generally consists of a generator and a discriminator, and is trained in an adversarial manner. GAN has many advantages, it can train any kind of generator network, and its design also does not need to follow any kind of factorization model, nor does it need to use Markov chains to repeatedly sample, and it does not need to infer during the learning process, but the GAN has the problem that the network is difficult to converge. Therefore, in [10,11], it is suggested that Wasserstein-1 distance and gradient penalty is used to improve the stability of the optimization process. Conditional GANs (cGANs) [2,12] take conditional variables as inputs to the generator and discriminator, to generate images with desired properties.

### 2.2. Image-to-Image Translation

Good results have been achieved in image-to-image style transfer research [12–15]. Pix2pix [12] uses cGAN [2] to train the model in a supervised manner, combining adversarial loss and L1 loss, so paired samples are required. In order to solve the problem that the data need to be paired, the unpaired image transformation framework has been proposed [13–15]. UNIT [14] proposes to add VAE [16] to CoGAN [17] for unsupervised image-to-image translation, which builds two encoders sharing the same latent space, and sharing weights, to learn the joint distribution of images across domains. CycleGAN [14], NICE-GAN [15], and DiscoGAN [13] preserve key properties between input and translated images by exploiting cycle consistency loss, but they can only learn the relationship between two different domains at a time. To address this problem, StarGAN [8] proposes the use of a generator that can generate images of all domains. Instead of just taking images as conditional input, StarGAN also takes the label of the target domain as input, and the generator is used to transform the input image to the target domain indicated by the input label. Moreover, DualStyleGAN [18] adds an external style control module on the basis of StyleGAN [19], and learns external styles on small-scale data through a progressive transfer learning method, which can effectively imitate the style of artistic portraits and achieve sample-based high-definition stylized faces. RAMT-GAN [20] realizes cross-domain image conversion based on a dual input/output network constructed by the BeautyGAN [21] architecture, and introduces identity preservation loss and background invariance loss to ensure that the generated facial makeup images are accurate and realistic. Different from the above methods, our framework not only uses a single model to learn the relationship between multiple domains, but also introduces an attention module to make the features of the generated images more obvious and improve the quality of the generated images.

## 3. The Proposed Method

In this section, we describe our proposed framework and its training objective function.

### 3.1. The Proposed Framework

X and Y represent the image set and the domain of the image, respectively. Given an image $x \in X$ and an arbitrary domain $y \in Y$, where y is the encoding of the field Y, our goal is to train a generator G, that can generate different images of different domains y corresponding to images x. The generator G adopts an encoder–decoder structure. Spatial pooling is necessary to extract high-level abstract representations of image features, but spatial pooling reduces the spatial resolution and fine details of the image feature map, and

the features map will easily lose details when it is restored later. In order to improve the quality of the coded image, some skip connections are applied between the encoder and the decoder to prevent the important features of the image from being lost. Skip connections are added between every two corresponding encoder layers and decoder layers. When the network is very deep, the decoder layer cannot complete the restoration of image details. The skip connections pass the information of the feature map through the convolutional layer to the corresponding part in the decoder layer. In addition, spatial attention and channel attention modules are applied in the encoder and decoder, so that the important features of the image are enhanced and the unimportant features are suppressed. Our framework mainly consists of three modules:

- Mapping network (F) and style encoder (E). The schematic diagram of their structure is shown in Figures 1 and 2 respectively. For a given latent code z and a domain y, or a given image x and the corresponding image domain y, the style code s = $F_y(z)$ generated by the mapping network, where $F_y(z)$ represents the F output of the corresponding domain y. The output of the style encoder is s = $E_y(x)$, where $E_y(x)$ represents the E output for the corresponding domain y. E consists of MLPs, convolutional neural net- works (CNNs) and residual blocks with multiple output branches, which can provide a variety of style encodings, and the number of output branches is determined by the number of image domains. F consists of multiple MLPs with output branches, and the number of output branches is also determined by the number of domains.

- Generator (G). As shown in Figure 3, this is a schematic diagram of our residual downsampling module, and Figure 4 is the schematic diagram of our AdaIN-residual upsampling module. Figure 5 is the generator structure diagram. The generator transforms an input image x into an output image G(x, s), where s is a domain-specific style code provided by a mapping network F or a style encoder E. Our generator consists of four downsampling blocks, two intermediate blocks and four upsampling blocks, all of which have pre-activated residual units. We apply adaptive normalization (AdaIN) [9,19] to the upsampling module in the generator, which can inject s into G. AdaIN receives two sources of information: the content input x and the style input s, and matches the channel-wise mean and standard deviation of x to the channel-wise mean and standard deviation of s. As shown in the following Equation (1). Simply speaking, AdaIN realizes style transfer by changing the data distribution of features at the feature map level, with small computational and storage costs, and is easy to implement, additionally, there are skip connections between the encoder and decoder, which can effectively avoid some important features of loss. In addition, the generator also adds the attention mechanism module (CBAM) [22] of the convolution module, to make the important features of the feature map more obvious and suppress the unimportant features.

$$AdaIN(x,s) = \sigma(s)\left(\frac{x - \mu(x)}{\sigma(x)}\right) + \mu(s) \qquad (1)$$

- Discriminator (D). Our discriminator D is a multi-task discriminator [23,24]. As shown in Figure 6. It contains multiple output branches, as well as multiple preactivated residual blocks. Each branch $D_y$ learns a binary classification to determine whether an image x is a real image of its domain y or a fake image G(x, s) generated by Generator.

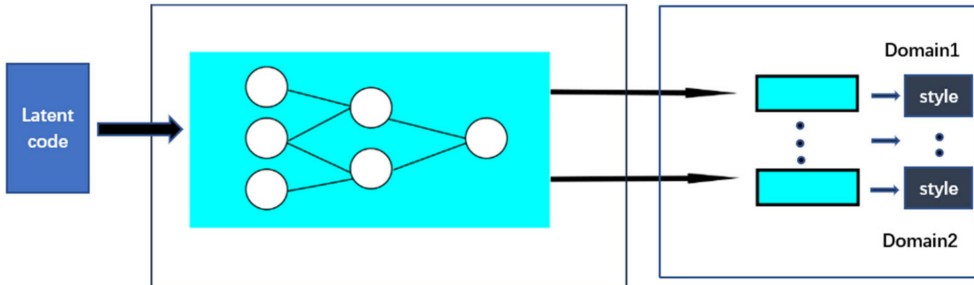

**Figure 1.** Mapping network.

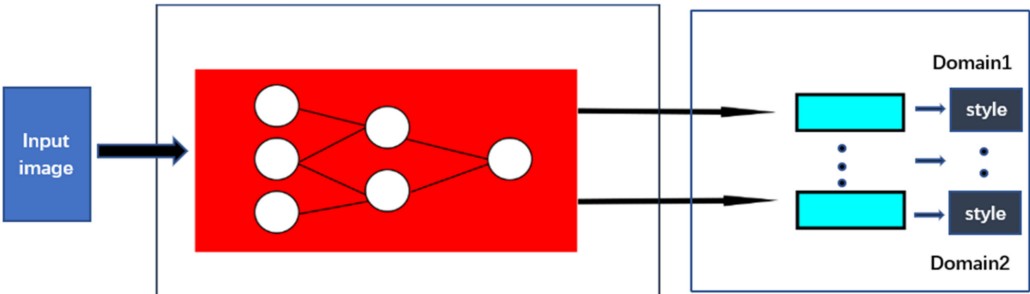

**Figure 2.** Style encoder.

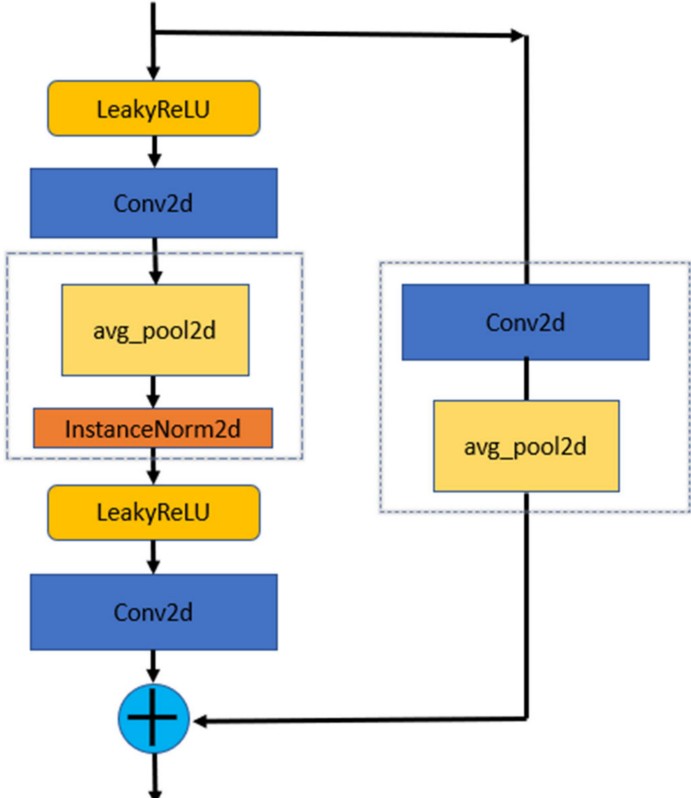

**Figure 3.** Residual block (the part within the dashed line indicates whether the judgment is executed or not).

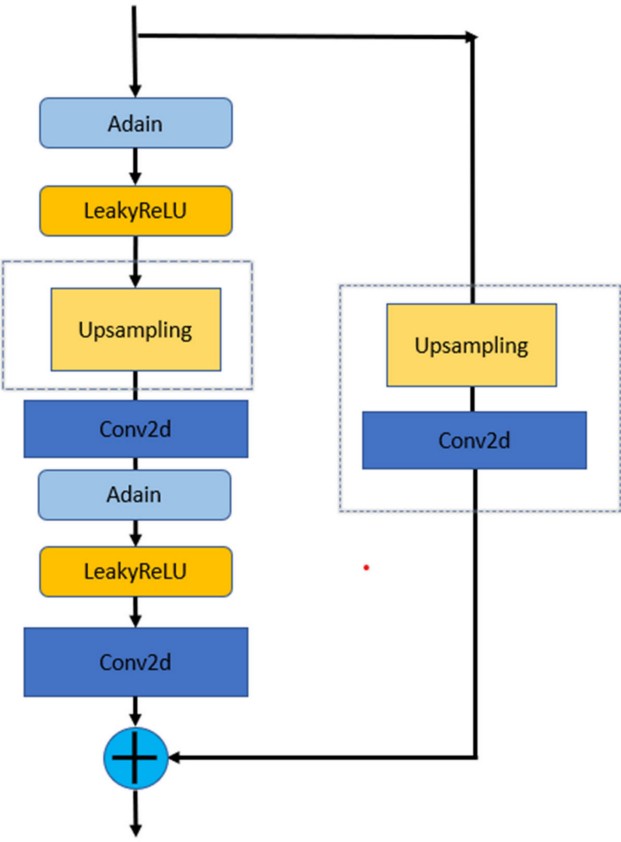

**Figure 4.** AdainResBlk block (the part within the dashed line indicates whether the judgment is executed or not).

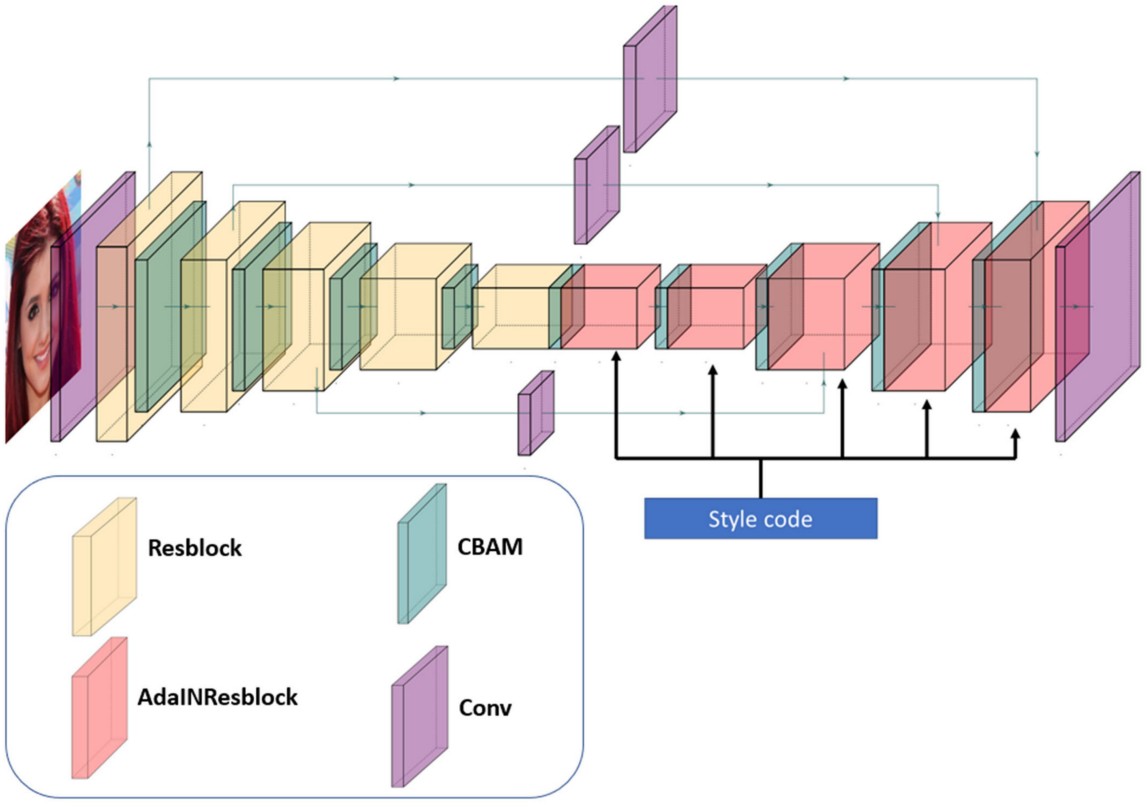

**Figure 5.** Generator.

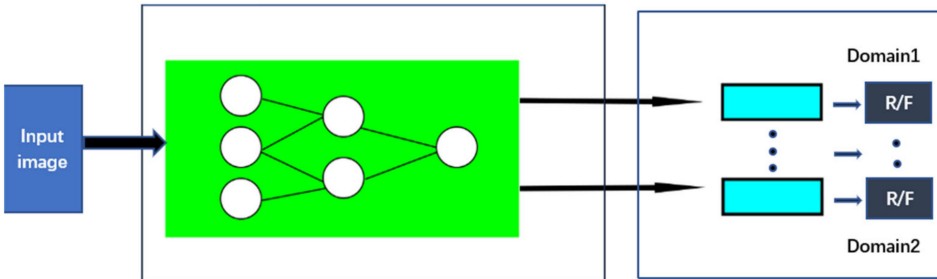

**Figure 6.** Discriminator.

*3.2. The Function of Training*

Our goal is to train a generator G that can learn mappings between multiple domains. For a given image x and y, we train as follows:

Adversarial loss: To make the generated image indistinguishable from the real image, an adversarial loss is used:

$$\mathcal{L}_{\text{adv}} = \mathbb{E}_{x,y}\left[\log D_y(x)\right] + \mathbb{E}_{x,\widetilde{y},z}\left[\log\left(1 - D_{\widetilde{y}}(G(x,s))\right)\right] \tag{2}$$

x is the input image, y is the image source domain, s is generated by the randomly sampled code $z \in Z$ and the image target domain $\widetilde{y} \in Y$ through the mapping network $s = F_{\widetilde{y}}(z)$ or style encoder $s = E_{\widetilde{y}}(x)$. The generator G takes the image x and the style code s as input to generate a picture G(x, s). Dy (·) represents the D output corresponding to the domain y. Generator scholars use s to generate images G(x, s) that are indistinguishable from real images of the domain $\widetilde{y}$.

Style reconstruction: In order for the generator G to use the style code when generating images and train a style encoder E to learn the output of different domains, the learned style encoder E allows G to transform the input image to reflect the style code of the reference image. Our style reconstruction loss is:

$$\mathcal{L}_{\text{sty}} = \mathbb{E}_{x,\widetilde{y},z}\left[\left\|s - E_{\widetilde{y}}(G(x,s))\right\|_1\right] \tag{3}$$

z is the latent code generated by random noise, $\widetilde{y}$ is the given image domain, x is the real picture, s is generated by z and $\widetilde{y}$ through the mapping network F or x and $\widetilde{y}$ through the style encoder E. Similar approaches have also been used by previous methods [25–27] with this loss. Most of them use multiple encoders to train different pictures to their latent codes; we only train one encoder to learn to map pictures of different domains to their latent codes.

Diverse styles: This loss is derived from MSGAN [28], regularizing the generator with a diversity-sensitive loss [28,29]:

$$\mathcal{L}_{\text{ds}} = \mathbb{E}_{x,\widetilde{y},z_1,z_2}[\|G(x,s_1) - G(x,s_2)\|_1] \tag{4}$$

where the style code $s_1$, $s_2$ is generated by two random latent codes $z_1$, $z_2$ through the mapping network $F\left(s_i = F_{\widetilde{y}}(z_i), i = 1,2\right)$. Compared with MSGAN, the denominator image is removed, making the training more stable. Maximizing regularization also enables the generator to discover more style features and generate diverse images.

Cycle consistency loss: This loss is derived from CycleGAN [6]. The purpose is to make the generated images properly maintain the characteristics of the original image:

$$\mathcal{L}_{\text{cyc}} = \mathbb{E}_{x,y,\widetilde{y},z}[\|x - G(G(x,s),\hat{s})\|_1] \tag{5}$$

Among them, where $\widetilde{s}$ is generated by the style encoder E from the source domain y corresponding to the input images x and x ($\widetilde{s} = E_y(x)$). By letting the generator G use

the style code s to reconstruct the input image x, the generator G can retain the original features of x when changing the style of x.

Total loss function: Our overall objective function is as follows:

$$\min_{G,F,E} \max_{D} \mathcal{L}_{adv} + \lambda_{sty}\mathcal{L}_{sty} - \lambda_{ds}\mathcal{L}_{ds} + \lambda_{cyc}\mathcal{L}_{cyc} \tag{6}$$

where $\lambda_{sty}$, $\lambda_{ds}$, $\lambda_{cyc}$ are the hyperparameters of each loss function. We train our model with the above objective function. We use in all experiments $\lambda_{cyc} = 1$, $\lambda_{sty} = 1$ and $\lambda_{ds} = 1$.

## 4. Experiments

In this section, we first compare recent image attribute transfer methods with our framework, through research. Next, we conduct classification experiments on image attribute transfer and synthesis. Finally, we show empirical results on image-to-image attribute transfer learned by our framework from several datasets. For different datasets, these models need to be trained separately for each dataset.

### 4.1. Baseline Model

We use MUNIT [25], DRIT [30] and MSGAN [28], as our baselines, all of which learn multimodal implicatures between two or more domains. For multi-domain comparisons, we train these models multiple times for the image domain.

MUNIT [25] reduces the image dimension of the dataset into two types of low-dimensional codes: content code and style code. It combines the content code with the style code of another image domain to generate style transfer and uses the decoder to increase the dimension of the newly combined code to generate the resulting image, before the generated image is decomposed into two codes again. For the original code to calculate the error back propagation, the c encoder and the s encoder should be well integrated in the decoder. Adaptive instance normalization (AdaIN) is used, along with an MLP network, which is used to generate parameters to assist residential blocks to generate high-quality images.

DRIT [30] proposes a decoupled representation-based method that can produce various outputs without paired training images. Embedding images into two spaces: an invariant domain content space that captures information shared across domains, and a domain-specific attribute space, using decoupled features as input, can greatly reduce mode collapse and achieve diversity. Introduce cross-cycle consistency loss to handle unpaired training data.

MSGAN [28] proposes an effective and simple pattern search regularization method that solves the pattern collapse problem of cGAN. The method is easy to add to existing models and has been proven to generalize well and effectively.

### 4.2. Dataset

We use CelebA-HQ [31] and the AFHQ dataset for experiments. We divide CelebA-HQ into two domains, male and female, and the AFHQ dataset is a dataset divided into three domains, cat, dog, and wildlife, each providing 5000 high-quality images at $512 \times 512$ resolution. We do not use additional information (other than domain labels), and learn information (such as styles) without supervision. For a fair comparison, we resize all images to a resolution of $256 \times 256$ for experiments.

### 4.3. Training

All models are trained using Adamw [32], $\beta = 0$, $\beta = 0.99$. For data addition, we simply crop the image and do some normalization. For all models, we set the learning rate to 1e-4 and the weight decay to le-4. Trained on 4 pieces of 2080Ti for about 3 days.

### 4.4. Evaluation Metrics

We use Frechet inception distance (FID) [33] to evaluate the quality of generated images, with lower scores indicating high correlation with higher-quality images; using learned perceptual image patch similarity (LPIPS) [34] to evaluate the diversity of generated images, with higher scores indicating better diversity of generated images.

### 4.5. Comparison of Image Synthesis

In this section, we evaluate the performance of the silent framework on image synthesis from two aspects: latent-guided synthesis and reference-guided synthesis.

**Latent-guided synthesis**. Figure 7 shows a qualitative comparison with related methods on the CelebA-HQ dataset. Each method uses random noise in the latent space to produce a diverse picture output. Figures 8 and 9 are qualitative comparisons on the AFHQ dataset.

Qualitative comparison of latent guided image synthesis results on CelebA-HQ and AFHQ datasets is undertaken. Each method uses a randomly sampled latent code to transform the source image (top row) into the target domain. To learn meaningful styles, we transform latent codes, z, into domain-specific style codes, s, through a mapping network, M. After injecting style codes into a generator, E, we use a style reconstruction loss that allows the generator to generate different images in field style.

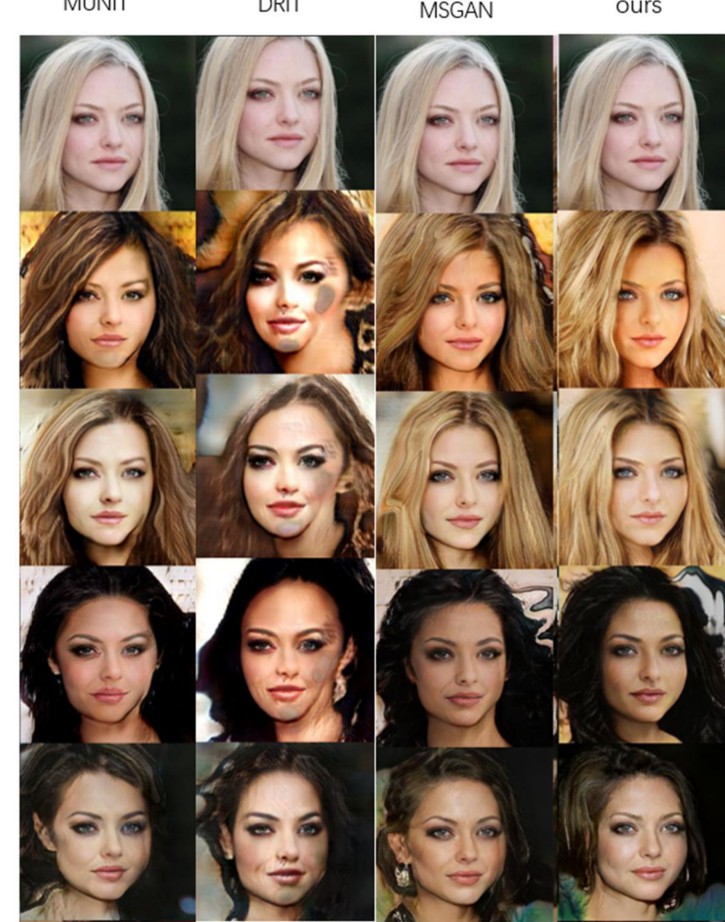

**Figure 7.** Using random noise to guide generation of images in CelebA-HQ.

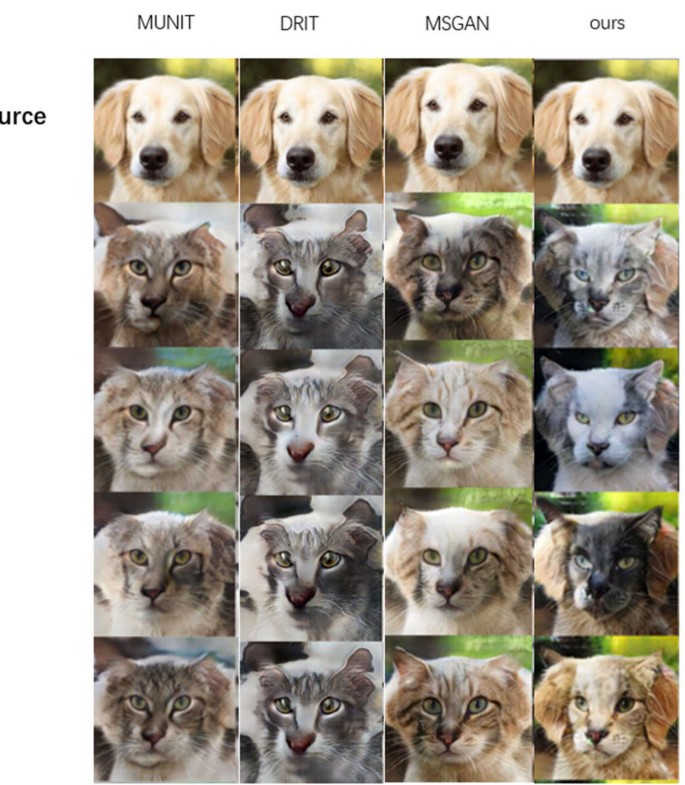

**Figure 8.** Using random noise to guide generation of images in AFHQ.

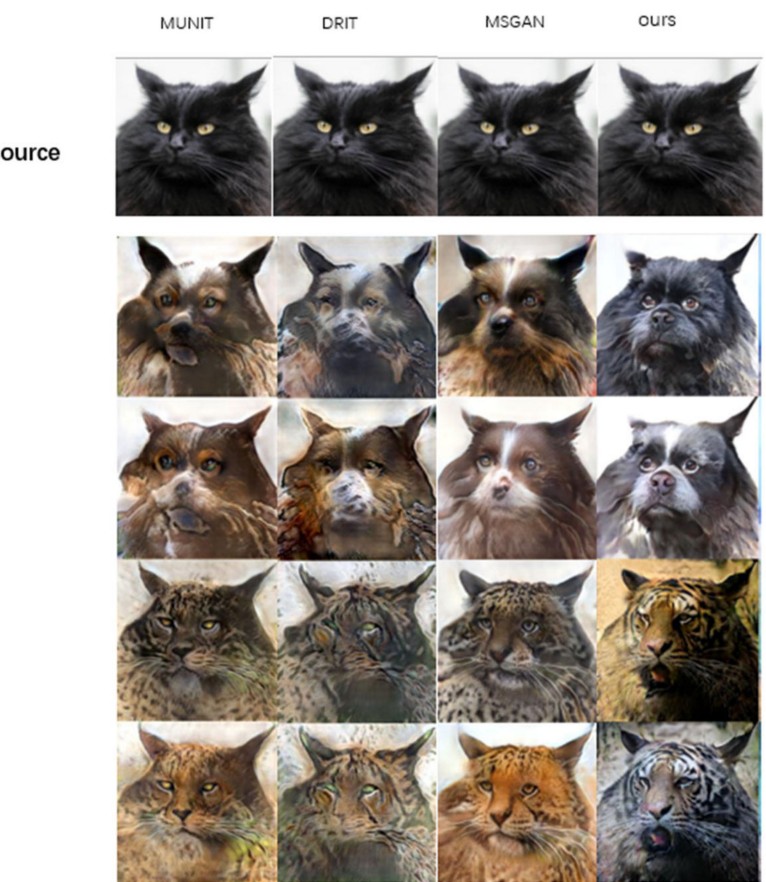

**Figure 9.** Using random noise to guide generation of images in AFHQ.

Table 1 shows that our method outperforms some methods. We outperform comparative methods in FID vs. LPIPS in CelebA-HQ dataset, but perform worse than the best comparison method MSGAN in dataset. We speculate that it may be that when the decoder adds attention mechanism to suppress the unimportant features of the image, it may also limit the development of image diversity, so that image diversity cannot generate pictures well, according to the provided style codes s.

**Table 1.** Quantitative comparison of latent-guided synthesis. (The bold number indicates the best result.)

| Method | CelebA-HQ | | AFHQ | |
|---|---|---|---|---|
| | **FID** | **LPIPS** | **FID** | **LPIPS** |
| MUNIT | 31.6 | 0.365 | 43.6 | 0.501 |
| DRIT | 52.3 | 0.176 | 95.4 | 0.328 |
| MSGAN | 33.5 | 0.375 | 61.6 | **0.517** |
| Ours | **18.6** | **0.423** | **28.6** | 0.412 |

**Reference-guided synthesis.** Figure 10 is the reference guide image synthesis result on CelebA-HQ. The source and reference images in the first row and column are real images, and the rest are images generated by our proposed model. Our model learns to transform source images that reflect the style of a given reference image, following high-level semantics, such as hairstyle, makeup, beard, and age, from the reference image, while preserving the pose and identity of the source image. Note that the images in each column share a logo with a different style, and the images in each row share a style with a different identity.

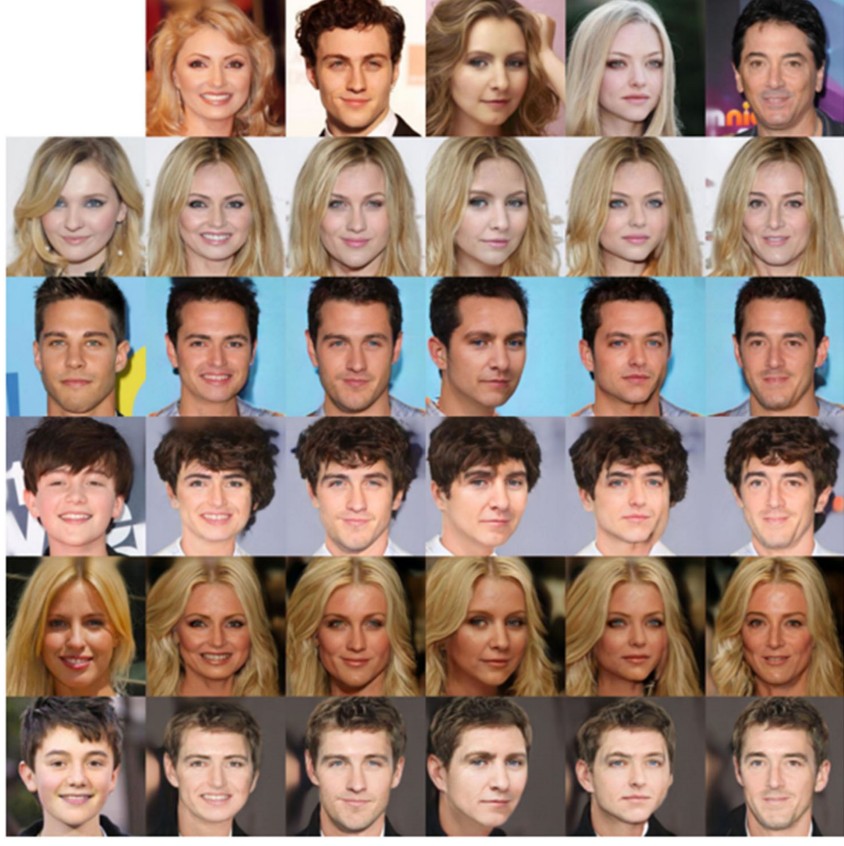

**Figure 10.** Reference guided image synthesis in CelebA-HQ (The first column is the original image; the first row is the reference image).

Figure 11 is a qualitative comparison of the synthetic results of reference guided images on the CelebA-HQ and AFHQ datasets. Each method transforms the source image into the target domain, reflecting the style of the reference image. The first column is the source image and the second column is the reference image.

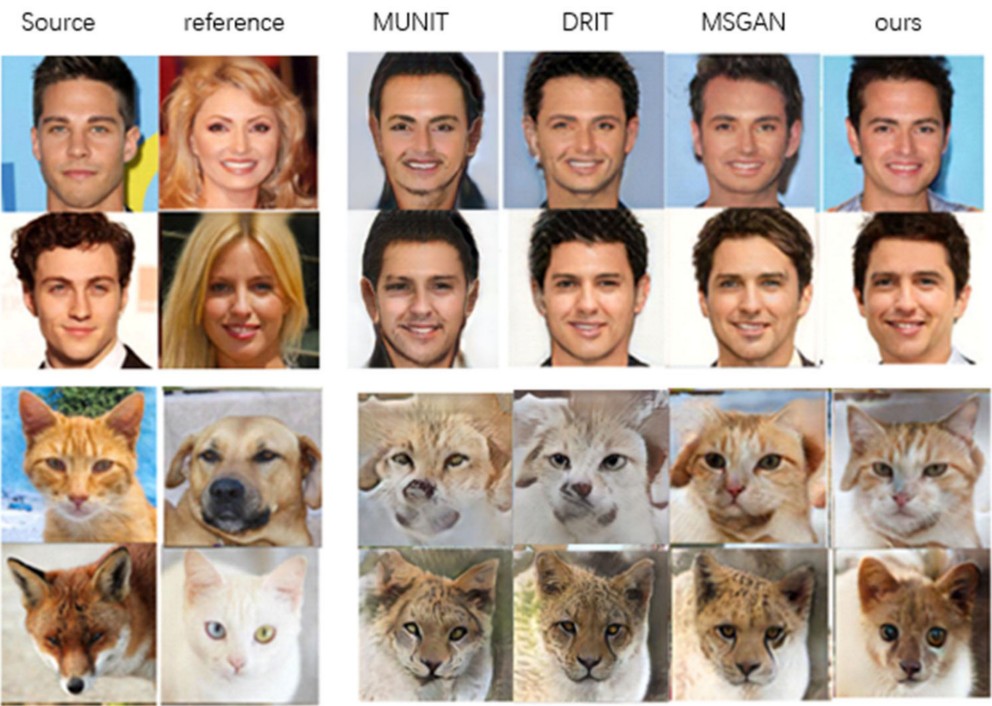

**Figure 11.** Comparison of reference guided syntheses.

Table 2 is a qualitative comparison with related methods, FID and LPIPS. When compared to these methods, ours performs the best; the images we generate are the best visually and have the most diversity of pictures.

**Table 2.** Quantitative comparison of reference-guided synthesis. (The bold number indicates the best result.)

| Method | CelebA-HQ | | AFHQ | |
|---|---|---|---|---|
| | **FID** | **LPIPS** | **FID** | **LPIPS** |
| MUNIT | 106.8 | 0.178 | 183.6 | 0.197 |
| DRIT | 53.4 | 0.311 | 114.4 | 0.192 |
| MSGAN | 38.9 | 0.324 | 68.7 | 0.159 |
| Ours | **28.1** | **0.382** | **26.6** | **0.399** |

We present some of our experimental results in the Appendix A. Figure A1 is the image generated using cycle consistency, Figure A2 shows the result of using random noise to guide generated images, and Figure A3 is the guided synthesis of images with reference images.

## 5. Conclusions

The framework we propose mainly solves the problem of converting images from one domain to different images in the target domain and supports multiple target domains in image conversion. The results show that our model can generate pictures of diverse styles in multiple domains, including some shown previously [25,28,30]. However, the quality of the generated images can be further improved and the diversity can be richer. In our design, the number of different domains does not affect the quality of the output and model

performance is not much different when using only a single-domain dataset compared to using a multi-domain dataset. In addition, the use of skip connections and CBAM attention modules can also make the generated images have higher visual quality, but we speculate that adding CBAM in the generator decoding part may affect the diversification of the generated images, so there is still much room for improvement. We hope that our work can be applied to the development of image translation programs in multiple domains.

**Author Contributions:** Funding acquisition, Z.Q.; Resources, Z.Q.; Supervision, Z.Q.; data curation, Z.Y.; Writing—original draft, Z.Y.; Writing—review and editing, Z.Y.; visualization, Z.Y. All authors have read and agreed to the published version of the manuscript.

**Funding:** This work was supported by Hainan Provincial Key Research and Development Program 361 (NO: ZDYF2020018), Hainan Provincial Natural Science Foundation of China (NO: 2019RC100), Haikou key research and development program (NO: 2020-049).

**Institutional Review Board Statement:** Not applicable.

**Informed Consent Statement:** Not applicable.

**Data Availability Statement:** The datasets used in this paper are public datasets. The CelebA-HQ could be found from https://drive.google.com/drive/folders/0B4qLcYyJmiz0TXY1NG02bzZVRGs (accessed on 22 March 2022). And the AFHQ could be found from https://github.com/clovaai/stargan-v2/blob/master/README.md#animal-faces-hq-dataset-afhq (accessed on 22 March 2022).

**Conflicts of Interest:** The authors declare no conflict of interest.

## Appendix A

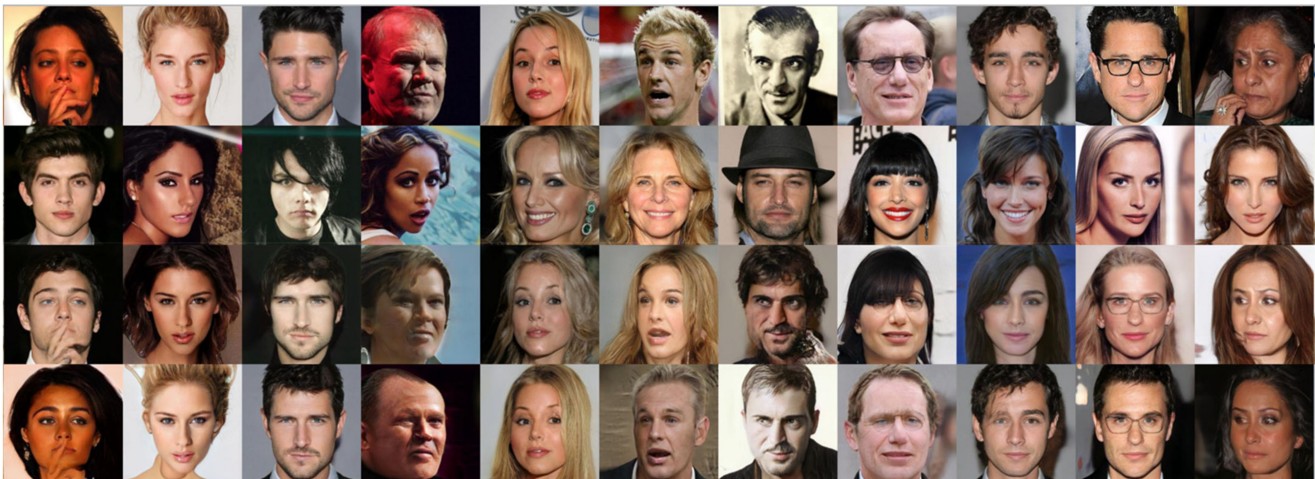

**Figure A1.** Images generated by cycle consistency (The first and second rows are real pictures).

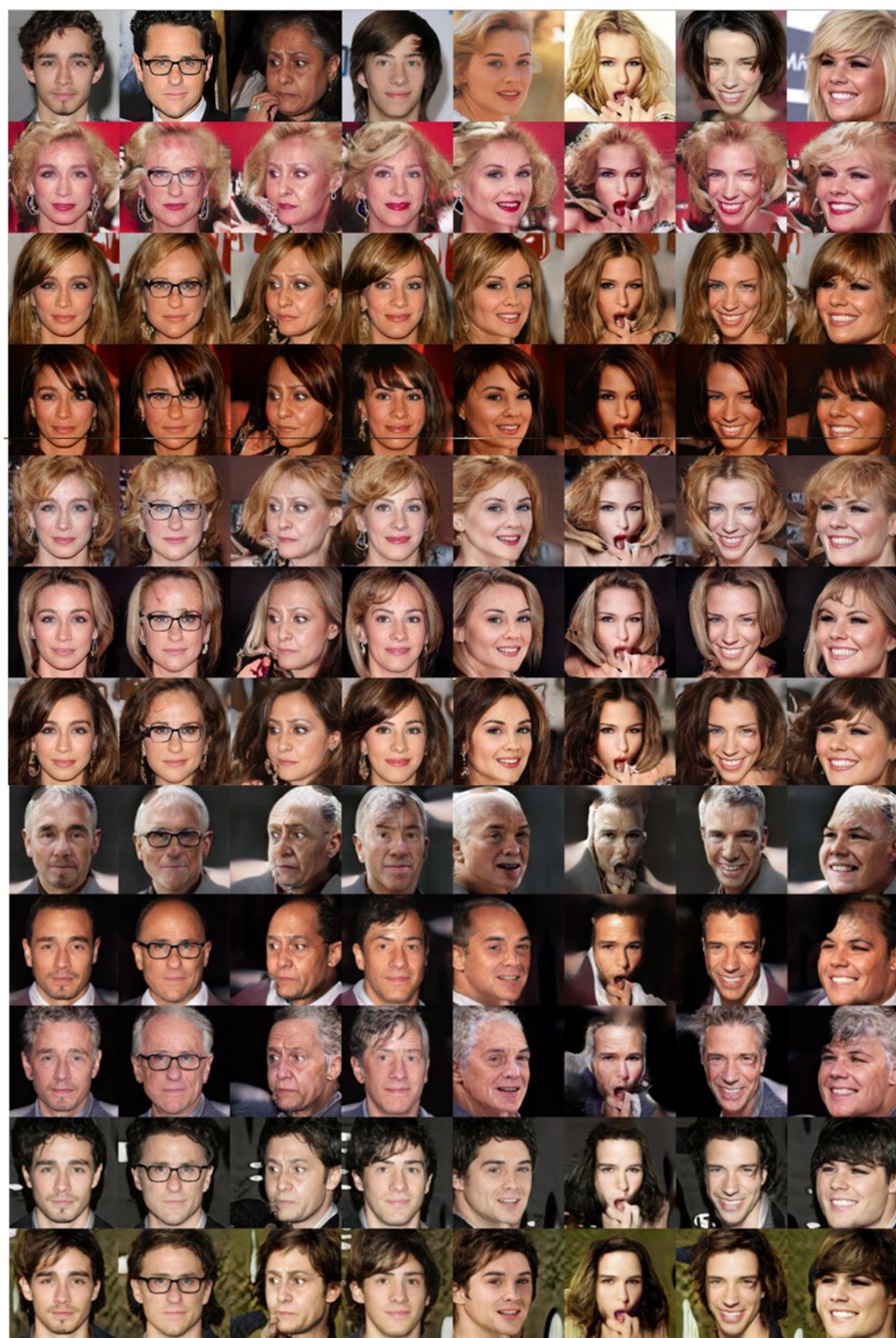

**Figure A2.** Using random noise to guide generated images s (The first row is the real picture).

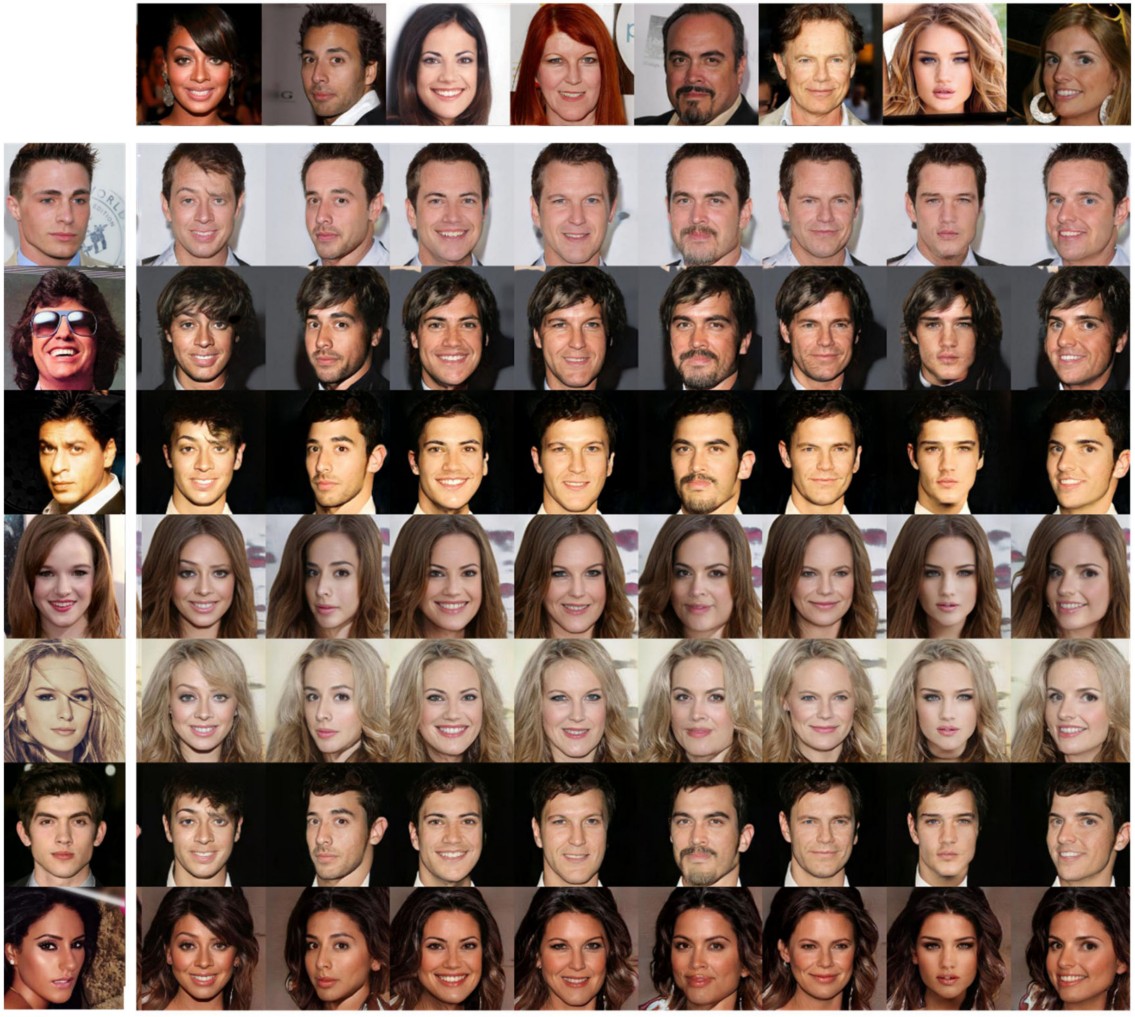

**Figure A3.** Reference guided synthesis images (The first column is the original image, the first row is the reference image).

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
