# Peer review of "An Image Style Diversified Synthesis Method Based on Generative Adversarial Networks"

_electronics, doi:10.3390/electronics11142235_

Round 1

Reviewer 1 Report

I have following general questions to the authors of the "An Image Style Diversified Synthesis Method Based on Generative Adversarial Networks" paper:

1) Based on paragraph from lines 201-203, could you better clarify if MUNIT, DRIT, MSGAN models were trained in the one go for both datasets or each dataset required separate training?

2) Is it possible to add any comments in the paper about scalability of your model? Like: How number of different domains affect the quality of output? What was your model performance in case of using only a single domain dataset? Was it the same as in a case with two domains?

3) How the values presented in Tables 1 and 2 were calculated? Were they averaged over set of generated images? (how many images were generated? more than presented examples in paper?)

I have also following specific remarks:

- lines 27-28: Because collecting ideal paired images with attributes that people want and don’t need is a formidable challenge... (doesn't make sense)

- line 31: unsupervised imageto-image compilation (lack of dash)

- line 60: generator network, Its design also (end of sentence?)

- line 104: three modules. as the picture shows (which picture? and also stray dot)

- line 126: transfer by chang ing the data (stray space)

- line 129: to implement. and there are skip (stray dot)

- Figures 1-3: Domian (Domain)

- Figure 5: architecture details in this picture are too small

- line 189: style code se to reconstruct (should be s or se here?)

- line 198: on image-toimage (lack of dash)

- line 210: is used. technology, while using (stray dot)

- line 234: Frecht inception distance (Frechet)

- line 236: patch similarity (LPIPS) [31]. to evaluate (stray dot)

- line 242: on the celeba dataset (better to use everywhere the same name of dataset - CelebA-HQ)

- Bibliography positions 14 and 15 are identical

Reviewer 2 Report

Style transfer is a widely researched topic today, with neural networks and many other techniques. GANs are one of the popular approaches used for this. In this sense, its important for the authors to highlight the differences with respect to the recent techniques.

Some of the papers that can be checked out for comparisons:

Kim YH, Nam SH, Hong SB, Park KR. GRA-GAN: Generative adversarial network for image style transfer of Gender, Race, and age. Expert Systems with Applications. 2022 Jul 15;198:116792.

Yang S, Jiang L, Liu Z, Loy CC. Pastiche Master: Exemplar-Based High-Resolution Portrait Style Transfer. InProceedings of the IEEE/CVF Conference on Computer Vision and Pattern Recognition 2022 (pp. 7693-7702).

Yuan QL, Zhang HL. RAMT-GAN: Realistic and accurate makeup transfer with generative adversarial network. Image and Vision Computing. 2022 Apr 1;120:104400.

Zhang T, Yu L, Tian S. CAMGAN: Combining attention mechanism generative adversarial networks for cartoon face style transfer. Journal of Intelligent & Fuzzy Systems. 2022 Jan 1(Preprint):1-9.

Ji F, Sun M, Qi X, Li Q, Sun Z. MOST-Net: A Memory Oriented Style Transfer Network for Face Sketch Synthesis. arXiv preprint arXiv:2202.03596. 2022 Feb 8.

Fang S, Duan M, Li K, Li K. Facial makeup transfer with GAN for different aging faces. Journal of Visual Communication and Image Representation. 2022 May 1;85:103464.

Guarnera L, Giudice O, Battiato S. Deepfake Style Transfer Mixture: A First Forensic Ballistics Study on Synthetic Images. InInternational Conference on Image Analysis and Processing 2022 (pp. 151-163). Springer, Cham.

In addition, authors can also report on the performance under the influence of noise in the test image used on the overall nature of style transfered. How much is the influence of the object background to the accuracy of the style transfer. What about the minimum resolution required and the resolution mismatches between the images. Some of these analysis is essential to benchmark the performance.

Also, the theme of special issue focuses on camera hardware - which is not reflected in this work. I do not think, in the current form it addresses any hardware specific issues.

Round 2

Reviewer 1 Report

I accept the authors response and have no further comments.